# Retrospectively Assessed Muscle Tone and Skin Colour following Airway Suctioning in Video-Recorded Infants Receiving Delivery Room Positive Pressure Ventilation

**DOI:** 10.3390/children10010166

**Published:** 2023-01-14

**Authors:** Gazmend Berisha, Anne Marthe Boldingh, Britt Nakstad, Elin Wahl Blakstad, Arild Erland Rønnestad, Anne Lee Solevåg

**Affiliations:** 1The Department of Paediatric and Adolescent Medicine, Akershus University Hospital, P.O. Box 1000, 1478 Lørenskog, Norway; 2Faculty of Medicine, Institute of Clinical Medicine, University of Oslo, P.O. Box 1171, 0318 Oslo, Norway; 3The Department of Anaesthesia and Intensive Care Unit, Stavanger University Hospital, P.O. Box 8100, 4068 Stavanger, Norway; 4Department of Paediatrics and Adolescent Health, University of Botswana, Private Bag, Gaborone 0022, Botswana; 5Department of Neonatal Intensive Care, Division of Paediatric and Adolescent Medicine, Oslo University Hospital, Rikshospitalet, Nydalen, P.O. Box 4950, 0424 Oslo, Norway

**Keywords:** neonatology, resuscitation, suction, clinical appearance, guideline compliance

## Abstract

Background: Recently, the International Liaison Committee on Resuscitation published a systematic review that concluded that routine suctioning of clear amniotic fluid in the delivery room might be associated with lower oxygen saturation (SpO_2_) and 10 min Apgar score. The aim of this study was to examine the effect of delivery room airway suctioning on the clinical appearance, including muscle tone and skin colour, of video-recorded term and preterm infants born through mainly clear amniotic fluid. Methods: This was a single-centre observational study using transcribed video recordings of neonatal stabilizations. All infants who received delivery room positive pressure ventilation (PPV) from August 2014 to November 2016 were included. The primary outcome was the effect of airway suctioning on muscle tone and skin colour (rated 0–2 according to the Apgar score), while the secondary outcome was the fraction of infants for whom airway suction preceded the initiation of PPV as a surrogate for “routine” airway suctioning. Results: Airway suctioning was performed in 159 out of 302 video recordings and stimulated a vigorous cry in 47 (29.6%) infants, resulting in improvements in muscle tone (*p* = 0.09) and skin colour (*p* < 0.001). In 43 (27.0%) infants, airway suctioning preceded the initiation of PPV. Conclusions: In this single-centre observational study, airway suctioning stimulated a vigorous cry with resulting improvements in muscle tone and skin colour. Airway suctioning was often performed prior to the initiation of PPV, indicating a practice of routine suctioning and guideline non-compliance.

## 1. Introduction

In 1971, Cordero and Hon [1] observed bradycardia in seven (15%) and apnoea in five (11%) out of forty-six infants who were subjected to oro- and nasopharyngeal suctioning with a 5 or 8 Fr feeding tube and a de Lee trap. Two infants were intubated, including one who also received chest compressions due to cardiac arrest [1]. Cordero and Hon were referenced in the American Heart Association (AHA)/International Liaison Committee on Resuscitation (ILCOR) 2000 guidelines [2,3], but it was not until 2010 [4] that AHA/ILCOR advised against routine oro- and nasopharyngeal suctioning because of concerns that the harm outweighs the potential benefits. A recent ILCOR systematic review [5] included nine randomized trials and two prospective observational studies and concluded that routine delivery room suctioning of clear amniotic fluid may be associated with a lower oxygen saturation (SpO_2_) at 5 min, an increased time to reach SpO_2_ of 86% and 92%, and a lower 10 min Apgar score. Serious adverse effects were not confirmed by the randomized trials in the ILCOR review. This may be due to selective reporting in the original studies or, possibly, recruitment bias, as the studies enrolled mostly healthy infants who were not in need of significant resuscitation. Furthermore, Cordero and Hon reported intrapartum oral and nasal suctioning followed by 10–20 s of blind suctioning after birth, a type of vigorous and repeated suctioning that is no longer common practice. However, it is still common to use airway suctioning as a corrective/optimising action when positive pressure ventilation (PPV) does not have the desired effects, i.e., when an obstructed airway is suspected [6].

In a delivery room observational study, we hypothesised that airway suctioning negatively affects infant clinical appearance and often precedes the initiation of PPV—a surrogate for being performed routinely. Furthermore, we hypothesised that suctioning is frequently performed despite the fact that national and international guidelines recommend against suctioning in the absence of airway obstruction. By including both vigorous and non-vigorous infants of different gestational ages, we hoped to contribute to filling the knowledge gap in our understanding of how delivery room airway suctioning may clinically affect different subgroups of infants.

## 2. Materials and Methods

A retrospective analysis of prospectively collected observational data from Akershus University Hospital (AUH) from August 2014 to November 2016 was performed. AUH has a 570,000-person catchment area and 5000 annual deliveries from 26 weeks of gestation onwards.

### 2.1. The Delivery Unit

All delivery rooms were equipped with a resuscitation table with radiant heat, a pulse oximeter, an oxygen blender, a gas-driven suction device, a self-inflating bag and a T-piece resuscitator. The equipment for endotracheal intubation and intravenous access was readily available, whereas electrocardiogram (ECG) monitoring was not routinely used in the data collection period. Airway suctioning was performed with a suction catheter with a maximum negative pressure of 100 mmHg (13 kPa, 135 cm H_2_O), as recommended in [7,8], while visible secretions outside of the mouth and on the face were wiped off with a towel. Midwives managed all stages of labour unless a complicated birth was foreseen. In such cases, one or more obstetricians were responsible. A paediatric resident was present at deliveries prior to 36 weeks of gestation or in the case of obstructed labour or suspected asphyxia. The Norwegian neonatal intensive care units (NICUs) did not have designated resuscitation teams or respiratory therapists. In the case of an infant in unforeseen poor condition at birth or a grade I caesarean section (grade I: immediate; grade II: fast; grade III: early delivery, but no maternal or foetal compromise; and grade IV: delivery at a time to suit the woman and maternity services), a paediatric resident and a consultant neonatologist or paediatrician were paged. The paging of NICU nurses was practised as needed.

All health personnel involved in neonatal resuscitation and stabilization followed the Norwegian Resuscitation Council (NRC) guidelines for neonatal resuscitation, adapted from ILCOR [9], the Australian and New Zealand Committee on Resuscitation [10] and the European Resuscitation Council [11]. The 2015 Norwegian guidelines recommend PPV with a T-piece or self-inflating bag for non-breathing or gasping newborns with a heart rate (HR) of <100 beats per minute (bpm) for 60 s before reassessing their HR. If their HR remains <100 bpm, another 30–60 s PPV should be provided before the initiation of chest compressions, which are performed if their HR is <60 bpm. Intubation was envisaged at several stages but was not mandatory before the initiation of chest compressions. It should be noted that the 2010 guidelines that were followed during parts of the data accumulation period recommended adequate PPV for ”at least 30 s” prior to considering chest compressions [4]. The NRC airway suction recommendations advise that ventilation must be started in all infants who need it without prior suctioning [12], i.e., routine suctioning is not recommended. Oro-pharyngeal suction may delay spontaneous respiration and the initiation of necessary PPV and cause laryngeal spasm, vasovagal bradycardia and oedema of the airways. Even suctioning meconium from the infant’s airways is not recommended as it does not reduce the incidence of meconium aspiration syndrome and can delay the start of PPV. Suctioning should only be performed if mucus, vernix, meconium or blood is suspected of obstructing the airway and should then be performed under visual guidance, ideally with a laryngoscope and a thick suction catheter [12].

### 2.2. Data Accumulation

Video cameras (VC) with audio (Hikvision 2 megapixel IP camera, Hangzhou, China) and motion-activation were installed under the resuscitation tables’ overhead warmers. VC focused on the infant and the health personnel’s hands on the infant, but not on the patient monitor or personnel present. All infants placed on the resuscitation table during the study period were video filmed and assessed for eligibility. Videos containing PPV were downloaded to a computer and reviewed. The inclusion criteria in this study incorporated all infants, regardless of their gestational age, who were brought to the resuscitation table and underwent stabilization and resuscitation actions, including PPV and airway suctioning. Non-suctioned infants who underwent PPV were included as a control group with regard to the immediate clinical outcomes.

### 2.3. Data Analysis and Refinement

A research assistant with no practical neonatal resuscitation experience and no knowledge of the study’s research questions transcribed the videos using Interact software version 9 (Mangold Int GmbH, Arnstorf, Germany). A paediatrician (AMB) made transcripts of a random selection of 20 of the recordings. Although no inter- or intra-rater variability test was performed, these 20 transcripts were considered to be consistent with the research assistant’s transcripts of the same recordings.

The transcripts contained information on airway suctioning, provision of PPV, intubation and chest compressions, spontaneous breathing, crying, muscle tone and skin colour. A vigorous infant was defined as one who breathed spontaneously, cried and had good muscle tone, the last of which was evaluated by flexed extremities and spontaneous movements.

This article presents the secondary processing of some transcripts presented in [13,14,15]. The primary use of the collected data was to scrutinize the effect of “high frequency, short duration” simulations and facilitator-led debriefings where the video logs were used for reflection and learning. ALS, a consultant neonatologist and GB, analysed the transcripts quantitatively and qualitatively.

Primary outcome:

Effect of airway suctioning on muscle tone and skin colour.

Secondary outcome:

The fraction of airway suctioning before the initiation of PPV as a surrogate for “routine” airway suctioning, i.e., guideline non-compliance.

Statistical analyses were performed in SPSS v28 (SPSS Inc., Chicago, IL, USA). Continuous data are presented as numbers with percentages or medians with interquartile ranges (IQR). Muscle tone and skin colour were scored 0–2, as in the Apgar score [16], immediately before and immediately after a suction episode. The McNemar test was used to analyse differences in the proportion of scores 0, 1 and 2 before and after airway suctioning. A *p*-value of < 0.05 was considered significant. Analyses were performed for the entire cohort, as well as for premature (gestational age (GA) ≤ 32 + 6 weeks) and late preterm/term (GA ≥ 33 + 0 weeks) infants born through clear and meconium-stained amniotic fluid. A cohort of non-suctioned infants receiving PPV was used as control. A comparison of the subgroups was performed with a Mann–Whitney *U* test for continuous variables and a Chi-Square test for categorical variables.

### 2.4. Ethical Considerations

The Regional Committee for Medical and Health Research Ethics South East approved the project. Video recording and evaluation were considered quality assurance and associated with minimal risk. Thus, the institutional review board at AUH approved presumed consent from the parents (reference 14-032). A written notification, including a withdrawal form for the study, was given to all women planning to give birth at AUH. The parents could bring the withdrawal form to the hospital as a way to opt out. They could also withdraw verbally and have the video deleted. A description of the study was found on a publicly available webpage. All health personnel were notified of the study and could refuse participation and have video recordings deleted without review. Due to confidentiality requirements, health personnel’s ages, experiences and training were not collected. The institutional review board’s approval and acceptance were dependent on the deletion of all videos after scrutiny and transcription.

## 3. Results

During the study period, 11,873 infants were delivered at AUH. Out of 397 video recordings, 315 were of good quality, but 13 transcripts were excluded due to technical failure (damaged files that would not open). Airway suctioning was evident in 160 transcripts, of which one was excluded because of missing information with regard to the resuscitation and stabilization measures.

Thus, 159/302 (52.6%) included transcripts containing episodes of airway suctioning, corresponding to 1.3% of all deliveries. Characteristics of suctioned (*n* = 159) and non-suctioned (*n* = 143) infants are presented in Table 1 (preterm and late preterm/term infants). Table 2 presents characteristics of the suction episodes in the 159 infants who received airway suctioning and PPV. The majority (90%) of the suctioned infants were delivered through clear amniotic fluid, and 29 (10%) were born before gestational week 33 + 0. We analysed 346 suction episodes with a median (IQR) duration of 11 (7–20) s. The median (IQR) number of suction episodes per infant was 2 (1–3), ranging up to 12 episodes in 1 infant.

A total of 226 suction episodes occurred in spontaneously breathing infants, whereas 120 episodes occurred in what appeared to be an obstructed airway. Fifteen (9.4%) of the infants were suctioned in a timely relationship to an intubation attempt. The infants with clear amniotic fluid accounted for 295/346 (85.3%) of the suction episodes in our material, whereas infants with meconium-stained amniotic fluid accounted for 51/346 (14.7%). This means that the infants with clear amniotic fluid were suctioned, on average, two times, whereas infants with meconium-stained amniotic fluid were suctioned three times on average.

### 3.1. Effect of Airway Suctioning on Muscle Tone and Skin Colour

Airway suctioning stimulated a vigorous cry in 47 (29.6%) infants, resulting in improvements in muscle tone (*p* = 0.09) and skin colour (*p* < 0.001).

As seen in Table 3, in the 85 (53.5%) infants that were vigorous from birth until the end of stabilization, airway suctioning used as the first measure or as a repeated measure did not affect muscle tone. In the 35 (22.0%) non-vigorous infants that were admitted to the NICU because spontaneous respiration remained inadequate, suctioning alone—and also suctioning combined with PPV and tactile stimulation—did not improve muscle tone. Nineteen (12.0%) infants were intubated because of insufficient ventilation and prolonged resuscitative actions but, presumably, not as a consequence of airway suctioning.

### 3.2. Airway Suctioning Prior to the Initiation of PPV

In the entire cohort, 43 (27.0%) infants had airway suctioning performed prior to PPV. In the remaining infants, suctioning was often performed at the very end of stabilization and often in infants described as vigorous.

### 3.3. Non-Suctioned Infants and Immediate Outcomes

All of the non-suctioned infants needed PPV to transition from intra- to extrauterine life. Fifteen (100%) preterm infants were admitted to the NICU and were still in the NICU for ongoing treatment 24 h after birth. Two infants (1.4%), one preterm and one term, needed intubation due to prematurity and prolonged PPV, respectively. Eight (6.3%) late preterm/term infants received chest compression; no deaths occurred in the group of non-suctioned infants.

**Table 1 children-10-00166-t001:** Characteristics and immediate outcomes of suctioned (*n* = 159) and non-suctioned (*n* = 143) preterm and late preterm/term infants who also received PPV.

	Suction GA ≤ 32 + 6 N = 29	No Suction GA ≤ 32 + 6 N = 15	*p*-Value	Suction GA ≥ 33 + 0 N = 130	No Suction GA ≥ 33 + 0 N = 128	*p*-Value
GA (weeks) median IQR	29 (28–31)	30 (29–31)	0.14	40 (38–41)	40 (39–41)	0.71
Female number (%)	10 (34.5)	6 (40.0)	0.75	60 (46.2)	56 (44.0)	0.70
BW (g) median IQR	1325 (1060–1674)	1255 (1075–1580)	0.99	3608 (3018–3973)	3517 (3062–3975)	0.71
Apgar 1 median IQR	6 (3–8)	6 (4–7)	0.71	5 (3–7)	6 (4–7)	0.037
Apgar 5 median IQR	8 (5–9)	8 (8–9)	0.16	8 (6–8)	9 (8–10)	<0.001
Apgar 10 median IQR	9 (8–10)	9 (9–10)	0.38	9 (8-10)	10 (9–10)	<0.001
ETI number (%)	9 (31.0)	1 (6.6)	0.12	10 (7.7)	1 (0.8)	0.010
Chest compression number (%)	4 (14.3)	0 (0.0)	0.28	20 (15.4)	8 (6.3)	0.026
Fetal HR number (%) Normal (Baseline HR 110–160 bpm) [17] Abnormal (*Tachycardia:* Baseline HR > 160 bpm for ≥ 10 min; *Bradycardia:* Baseline HR < 110 bpm for ≥10 min; or *Variable/late prolonged decelerations)* [17] Not measured	9 (31.0) 6 (20.7) 14 (48.3)	5 (33.0) 5 (33.0) 5 (33.0)	0.56	74 (57.8) 37 (28.9) 17 (13.3)	71 (56.0) 31 (24.0) 26 (20.0)	0.27
Delivery mode number (%) Vaginal Acute CS Elective CS	6 (20.7) 23 (79.3) 0 (0.0)	1 (7.0) 12 (80.0) 2 (13.0)	0.079	80 (62.5) 44 (34.4) 4 (3.1)	84 (66.0) 37 (29.0) 6 (5.0)	0.59
Presentation number (%) Cephalic Breech Transverse	15 (51.7) 12 (41.4) 2 (6.9)	10 (67.0) 4 (27.0) 1 (7.0)	0.61	99 (77.3) 24 (18.8) 5 (3.9)	105 (82.0) 23 (18.0) 0 (0.0)	0.079
Infant postresc. number (%) Normal (given to their mother) NICU (admission for further treatment) Died in DR or OR	1 (3.4) 27 (93.1) 1 (3.4)	0 (0) 15 (100) 0 (0.0)	0.58	53 (41.4) 74 (57.8) 1 (0.8)	96 (75.0) 32 (25.0) 0 (0.0)	<0.001
HIE and TH number (%)	0 (0.0)	0 (0.0)	N/A	5 (3.8)	1 (1.0)	0.21
Outcome 24 h number (%) Normal (given to their mother) NICU (still in need of further treatment) Died in NICU	1 (3.4) 27 (93.1) 1 (3.4)	1 (7.0) 14 (93.0) 0 (0.0)	0.69	65 (50.8) 62 (48.4) 1 (0.8)	95 (74.0) 33 (26.0) 0 (0.0)	<0.001
Dead before HDC number (%)	1 (3.4)	0 (0.0)	0.54	1 (0.8)	0 (0.0)	0.49

GA—Gestational age. IQR—Interquartile range. PPV—Positive pressure ventilation. Bpm—Beats per minute. HR—Heart rate. Postresc—Post-resuscitation. CS—Cesarean section. NICU—Neonatal intensive care unit. OR—Operating room. DR—Delivery room. HDC—Hospital discharge. ETI—Endotracheal intubation. CC—Chest compression. N/A—not applicable.

**Table 2 children-10-00166-t002:** Characteristics of suction episodes in 159 preterm and late preterm/term infants who received airway suctioning and positive pressure ventilation.

	Preterm GA < 32 + 6 N Infants = 29 N Suction Episodes = 66	Term GA > 33 + 0 N Infants = 130 N Suction Episodes = 280	*p*-Value
Suction before PPV infants number (%)	6 (20.7)	37 (28.5)	0.49
Suction CLAF episodes number (%)	61 (92.4)	234 (83.6)	0.082
Suction MSAF episodes number (%)	5 (7.6)	46 (16.4)	0.082
Suction Depth episodes number (%) Superficial (nose, mouth) Deep (trachea, in endotracheal tube) Superficial plus deep	20 (30.3) 7 (10.6) 39 (59.1)	146 (52.1) 41 (14.7) 93 (33.2)	<0.001

GA—Gestational age. PPV—Positive pressure ventilation. CLAF—Clear amniotic fluid. MSAF—Meconium-stained amniotic fluid.

**Table 3 children-10-00166-t003:** Effect of airway suctioning on muscle tone and skin colour.

	Preterm GA ≤ 32 + 6 N%	Late Preterm/Term GA ≥ 33 + 0 N%	Preterm GA ≤ 32 + 6 N%	Late Preterm/Term GA ≥ 33 + 0 N%
Airway Suction Effect	Vigorous (*n* = 18)	Vigorous (*n* = 67)	Non-vigorous (*n* = 15)	Non-vigorous (*n* = 67)
A vigorous cry with improved muscle tone and skin colour	N/A	0 (0.0)	3 (1.88)	44 (27.7)
Single/repeated airway suctioning in vigorous infants without affection of muscle tone	18 (11.3)	67 (42.1)	N/A	N/A
Persisting inadequate respiration in infants admitted to NICU without improvement in muscle tone when airway suctioned and/or suctioning combined with PPV + tactile stimulation	N/A	N/A	12 (7.5)	23 (14.5)
Intubated infants when airway suctioning performed during resuscitation *	N/A	N/A	9 (5.7)	10 (6.3)

GA—Gestational age. NICU—neonatal intensive care unit. PPV—positive pressure ventilation. N/A—not applicable. * A total of 15 infants were suctioned in a timely relationship to an intubation attempt, while the remaining 4 infants were suctioned before intubation was tried but not in relation to the intubation attempt itself.

## 4. Discussion

In this retrospective assessment of prospectively collected observational data, half of the infants who required delivery room PPV had their airways suctioned, confirming our hypothesis that airway suctioning is frequently performed despite national and international guidelines recommending the opposite. To address the knowledge gaps identified by ILCOR, we included both vigorous and non-vigorous (i.e., at a higher risk of needing resuscitation) premature and late preterm/term infants, mainly born through clear amniotic fluid. We found that airway suctioning often elicited a vigorous cry and subsequently improved the infants’ clinical appearance as assessed by muscle tone and skin colour.

Suctioning was often performed multiple times, especially in infants born through meconium-stained amniotic fluid. Airway suction preceded the initiation of PPV in almost one-third of the cases. The immediate improvement in muscle tone and skin colour following the suctioning of initially depressed or non-vigorous infants is a striking finding. However, the reliability of transcribed muscle tone and skin colour may be low. Thus, since the data are observational and not the results of an adequately powered randomized controlled trial, these results should be interpreted with care.

Routine delivery room airway suctioning is not recommended [18,19,20,21,22,23]. However, the widely referenced observational study that showed that oro- and nasopharyngeal suctioning caused vagally induced bradycardia and apnoea [1] is 50 years old. The recent ILCOR review and the included randomized controlled trials did not confirm these findings, even when high negative pressure was applied [4]. As many as 15% of the infants in our study received chest compressions, but probably not as a direct result of suctioning.

Comparing our study to the individual studies included in the ILCOR review, there are some important differences. Two out of the eleven studies compared oro-/nasopharyngeal suctioning using a bulb syringe to no suctioning [24,25]. The studies included term infants born only vaginally [24,25,26,27,28,29], only by caesarean section [30,31] or both vaginally and by caesarean section [32]. Our study includes both preterm and term infants, born vaginally and by caesarean section, while in ILCOR’s review, only Konstantelos et al. [31] and Kelleher et al. [24] included both preterm and term infants. Thus, a significant limitation of the studies included in the review [5] is the non-inclusion of preterm infants, maintaining a knowledge gap with regard to the role of suctioning in this population. Premature infants are more likely to need PPV, might not be able to clear their airway secretions as adequately as late preterm and term infants do, and may thus need airway suctioning. The studies included either infants born only through clear amniotic fluid [21,24,25,26,27], only meconium-stained amniotic fluid [23] or both meconium-stained and clear amniotic fluid [22,28]. Our study included infants born through mainly clear amniotic fluid. The outcome variables in our study are different from those of the eleven studies included in the ILCOR review, as we only included clinical—not physiological—variables.

International resuscitation guidelines recommend the considerate use of oropharyngeal suctioning. According to the neonatal resuscitation program (NRP), suctioning should only be considered if PPV is inefficient after tightening of the mask and repositioning of the head. Recently, ILCOR’s systematic review on clear amniotic fluid suctioning highlighted the need for more studies on non-vigorous, as well as premature infants [5], but concluded that suctioning of clear amniotic fluid from the upper airway (nose, mouth) should not be used as a routine option for newborn infants at birth (weak recommendation, very low certainty of evidence). However, airway readjustment and suction should be considered if airway obstruction is suspected (good practice statement) [5].

The limitations of this study include its observational and single-centre nature and the relatively small number of infants. The risk of bias in observational studies is high, and the analysis was performed based on transcripts as the source records were deleted. However, some of the videos were transcribed twice, with high similarities between the transcripts. Airway suctioning was performed based on subjective decisions by the patients’ caregivers. The appropriateness of interventions cannot be established with certainty based on the videos and their respective transcripts, as visual cues and clinical assessments are not always available for review. Importantly, the fact that our cohort only included infants receiving PPV may have biased our results. Due to confidentiality concerns, we did not examine the clinical experience of the team members in relation to their performance.

In this retrospective observational study of delivery room PPV, a very high fraction (49.5%) of infants had their airways suctioned, confirming our hypothesis that airway suctioning is frequently performed despite national and international guidelines recommending the opposite. To address the knowledge gaps identified by ILCOR, we included both vigorous and non-vigorous premature and term infants, mainly born through clear amniotic fluid. We found that airway suctioning often elicited a vigorous cry and subsequently improved the infants’ clinical appearance as assessed by muscle tone and skin colour.

There was frequent usage of repeated brief episodes of suctioning, especially in infants born through meconium-stained amniotic fluid. Airway suctioning preceded the initiation of PPV in almost one-third of the cases. The immediate improvement in muscle tone and skin colour following the suctioning of initially depressed or non-vigorous infants is a striking finding. However, the reliability of transcribed muscle tone and skin colour may be low. Thus, since the data are observational and not the results of an adequately powered randomized controlled trial, these results should be interpreted with care.

## 5. Conclusions

In some initially depressed infants in a Norwegian delivery unit, airway suctioning stimulated a vigorous cry with resulting immediate improvements in muscle tone and skin colour. Airway suctioning was often performed prior to the initiation of PPV.

## Data Availability

The data presented in this study are available from the corresponding author upon reasonable request. The data are not publicly available due to privacy restrictions.

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
