# Peer review of "Retrospectively Assessed Muscle Tone and Skin Colour following Airway Suctioning in Video-Recorded Infants Receiving Delivery Room Positive Pressure Ventilation"

_children, 2023, doi:10.3390/children10010166_

Round 1

Reviewer 1 Report

This paper describes a single centre observational study that examined video recordings of neonatal care in the minutes after birth with the aim of assessing the effect of upper airway suctioning prior to initiation of intermittent positive pressure ventilation on muscle tone and skin colour and the proportion of infants who received suctioning of nose and throat.  The authors acknowledge in the discussion an overall risk of bias. The study has the advantage of being a time-based cohort which should reduce risk of selection bias. Performance bias is likely to have been low because of the study was retrospective - decisions of the clinicians caring for each infant were unlikely to have been influenced by the study protocol (although if they did have prior awareness, this should be stated). It is not stated whether the research assistant who transcribed the videos and extracted data was aware of the purpose of the study and so there would be some concerns about observer bias. The methods section does not contain quite enough detail to know which analyses were decided a priori and which were post-hoc decisions, which could affect the risk of bias for some of the outcomes of the study.

Title: Ideally the title should include mention that this was a retrospective review of video-recordings.

Abstract:

1.      In a few minor aspects, the abstract could be better harmonised with the text.

2.      “…predominantly clear amniotic fluid…” is ambiguous – not obvious whether it means “nearly clear” or “in most cases, clear”.

3.      The proportions of infants receiving suctioning are hard to interpret without more consistent information about inclusion criteria (see comment in methods) and proportion of infants who met those criteria who were actually included.

Introduction :

4.      There are a few non sequiturs or possible errors of logic in the introduction, such as:

a.       page 1 line 42 – How is it expected that suctioning of the nose and throat would facilitate lung liquid clearance? Presumably the purpose of suctioning the nose and mouth/throat is to remove fluid in the upper airways, which is likely to be a mixture of amniotic fluid, oral secretions, and lung liquid? By 2010 it was well established that most lung liquid cleared after birth is removed across alveolar epithelia, and suctioning of the upper airway is presumably unlikely to influence this.

b.      On page 2, lines 49-50, it is not clear how “the risk of bias in non-randomised observations” would have reduced the sensitivity of the ILCOR systematic review to detect serious adverse effects. The lack of SAEs detected by the review is more likely to be due to selective reporting in the original studies, or possibly recruitment bias (the studies enrolled mostly healthy infants who were not in need of significant resuscitation).

5.      The abstract states that the effect of suctioning on tone and skin colour was the ‘main’ (presumably primary) outcome of the study and the rate (as a proportion) of babies receiving suctioning was a secondary outcome. It would be helpful to use the same ordering and identification of primary and secondary in the last paragraph of the introduction (which explains the hypotheses, but in reverse order).

Methods:

6.      The study is described as prospective, but the impression from section 2.3 is that this was a retrospective analysis nested within a prospective study.  Clarification would be helpful.

7.      The methods for the subgroup analysis specified in the last sentence of the introduction are not explained.

8.      Page 3 line 99 “All infants placed on the resuscitation table during the study period were recorded and assessed for eligibility….”. It would be helpful to the reader to set out the inclusion and exclusion criteria more clearly in the methods. There is an implication in the results that the only exclusions among infants placed on the resuscitation table were those whose parents opted out and those for whom the recording or transcript was uninterpretable, but is this correct? The abstract suggests that receiving positive pressure ventilation was also an inclusion criterion, but this is not clarified in the methods. The proportion of infants for whom recordings were made is only 3.3% of total births, which, if it resembles the number who were transferred to a resuscitation table, (or even the proportion receiving PPV) seems a surprisingly low proportion. Was there a high proportion of parents who opted out, or some other reason why there were so few eligible recordings?

9.      For the primary outcome - could the authors comment on why the decision was made to use skin colour judged from video recordings as an outcome? A previous study (O'Donnell CP, et al. Clinical assessment of infant colour at delivery. Arch Dis Child Fetal Neonatal Ed. 2007 Nov;92(6):F465-7) suggests) that there is very wide variation in SpO2 levels at which different observers consider that cyanosis has resolved, when observing video recordings. Does this observation have much value at all?  Was there a reason to not use Spo2 levels for the study instead of visual impressions?

10.   Also, how well can muscle tone be interpreted on a video recording? Do the authors have any evidence of the validity of judgements made from observation of videos, compared to the judgement of a person caring for the baby, (who can feel resistance to passive movements, test limb recoil etc. and assess reflex responses by handling the baby)?  Was it in fact only posture and responses to various kinds of stimuli that could be observed and how reliable are these observations?

11.   Why were other aspects of the Apgar score which may have greater reliability for reflecting the infant’s true state (such as heart rate) not assessed?

12.   For the secondary outcome, I recommend that more justification is needed as to why airway suctioning before initiation of PPV was regarded as evidence of guideline non-compliance. This may reflect the Norwegian newborn resuscitation guidelines, but many international guidelines do indicate that suctioning may be appropriate in the setting of evidence of obstruction to either spontaneous or assisted ventilation. Perhaps consider explaining the exact wording of the Norwegian guidelines to help explain the use of the proxy. While other aspects of the local guidelines are outlined in section 2.1, the criteria for suctioning are not specified.

13.   The term ‘ad modum’ may not be familiar to many readers – although it is a Latin term that can be appropriately translated to “consistent with” or “in the manner of” it is not in very common use. I suggest replacing with a more familiar term.

14.   Was use of control groups considered – e.g. observations of infants (at a comparable time point) who did not require respiratory support or did not receive suctioning?

Results:

15.   Page 4 line 155 – the implication is this sentence seems to be that if there was spontaneous breathing, the airway could not have been partly obstructed, or that suctioning is only appropriate in the setting of a complete obstruction. Is this implication intended? If so, is it justifiable?  

16.   Page 4, section on Effect of airway suctioning on muscle tone and skin colour – new subgroup analyses that were not mentioned in the introduction or methods are described. Please state in the methods whether these analyses were decided a priori or performed post-hoc. The same applies to the sub-analyses by gestation in Tables 1 and 2.

17.   Page 4, same section – The meaning of the statement that 19 infants were tracheally intubated, unrelated to whether suctioning was performed or not is not at clear. Please reword.

18.   In several places -- “who” is the usual relative pronoun used to describe human beings, not “that”.

19.   In Tables 1 and 2, what does “baby off crib number” mean? Please also explain the terms “normal”, “NICU” and “died or OR”? Are these categories meant to be mutually exclusive, and if so, why?

Discussion:

20.   Page 7, paragraph beginning line 237 – the implication is that the proportion of infants receiving suctioning are too high, but do we know what the ‘right’ rate is? Are the authors aware of any literature that reports what proportion of infants (and particularly what proportion of infants who are deemed to need some form of resuscitation) should optimally receive suctioning of the nose and throat? It is my understanding that the ILCOR treatment recommendations suggest against routine suctioning of all or most healthy infants but were careful that to acknowledge that the procedure would be sometimes required. Plugs of mucous, blood clots etc. (and in MSAF-exposed infants, clumps of meconium) can sometimes obstruct the airway, and these may not be revealed until the procedure is performed. As alluded to in the discussion, in an infant who is too preterm or too depressed to swallow effectively, even clear fluid could obstruct the airway. As indicated in the paragraph, the NRP course indicates airway suctioning as one of the potential “MR SOPA” ventilation-corrective steps. Before intubation, one would assume it is reasonable to suction a higher proportion of infants than those found to have something causing obstruction, on the basis that it would be less forgivable to miss a true obstruction.

21.   Following from the previous point, it is hard to discern whether the authors considered the rate (as a proportion) of suctioning in their study was optimal, too high, or too low, especially since they report that a third of infants improved after the procedure. Is it possible to draw any conclusion from this study? Regarding safety, without any data on heart rate or saturation levels, is it possible to draw any sound conclusions about safety from this study? What gap in knowledge does the study fill?

22.   Page 7, line 240, the ILCOR scoping review would have been superseded by the 2022 systematic review, so may not require citation. The consensus on gaps in knowledge identified by the systematic review should be readily available in the Consensus on Science with Treatment Recommendations from the systematic review (reference 4).

References:

23.   References 5  and 9 require correction of formatting and abbreviation style, and some of the references that are only available on-line have missing URL data.  

Author Response

Dear Editor of Children and the reviewers – thank you for valuable comments and suggestions for improvement of our original paper entitled “Retrospectively assessed muscle tone and skin colour following airway suctioning in video-recorded infants receiving delivery room positive pressure ventilation” (note: title revised after review).

We have revised the paper according to the suggestions and made a point-by-point description of the changes as well as responses/clarifications in the following. We hope that the manuscript has improved. 

Comments and Suggestions for Authors

This paper describes a single centre observational study that examined video recordings of neonatal care in the minutes after birth with the aim of assessing the effect of upper airway suctioning prior to initiation of intermittent positive pressure ventilation on muscle tone and skin colour and the proportion of infants who received suctioning of nose and throat. The authors acknowledge in the discussion an overall risk of bias. The study has the advantage of being a time-based cohort which should reduce risk of selection bias. Performance bias is likely to have been low because of the study was retrospective - decisions of the clinicians caring for each infant were unlikely to have been influenced by the study protocol (although if they did have prior awareness, this should be stated). It is not stated whether the research assistant who transcribed the videos and extracted data was aware of the purpose of the study and so there would be some concerns about observer bias. The methods section does not contain quite enough detail to know which analyses were decided a priori and which were post-hoc decisions, which could affect the risk of bias for some of the outcomes of the study.

Response: Thank you for your comment. We have revised the methods section to read: “A research assistant with no practical neonatal resuscitation experience and no knowledge of the study research questions transcribed the videos”

As stated in the methods section, the primary purpose of the video recording of resuscitations/stabilizations was "to scrutinize the effect of “high frequency, short duration” simulations with facilitator-led debriefings where the video logs were used for reflection and learning”. Thus, no specific research hypotheses were decided a priori, but emerged with time as we saw the richness of the transcripts and the opportunity to use the data to fill general knowledge gaps about neonatal resuscitation/stabilization.

Title: Ideally the title should include mention that this was a retrospective review of video-recordings.

Response: The title has been revised to: “Retrospectively assessment of muscle tone and skin colour following airway suctioning in video-recorded infants receiving delivery room positive pressure ventilation”.

Abstract:

  1. In a few minor aspects, the abstract could be better harmonised with the text.

Response: We have revised the abstract.

  1. “…predominantly clear amniotic fluid…” is ambiguous – not obvious whether it means “nearly clear” or “in most cases, clear”.

Response: We agree, the word predominantly has been replaced with the less ambiguous “mainly”.

  1. The proportions of infants receiving suctioning are hard to interpret without more consistent information about inclusion criteria (see comment in methods) and proportion of infants who met those criteria who were actually included.

Response: Inclusion criteria are all infants brought to the resuscitation table undergoing stabilization and resuscitation actions with the minimum actions being airway suction and PPV in any gestational age. The number of infants who met the inclusion criteria is 160, with one infant being excluded due to technical failure (the file is damaged and will not open). Four hundred video logs in total make up the accumulated database, but only 302 resuscitation cases were eligible for review. 159 (50,47%) out of the 302 video recordings contained airway suctioning and PPV. Looking at the revised table 1 You see that there were an additional 143 cases of non-suctioned infants who received PPV.

Introduction:

  1. There are a few non sequiturs or possible errors of logic in the introduction, such as:
    a) page 1 line 42 – How is it expected that suctioning of the nose and throat would facilitate lung liquid clearance? Presumably the purpose of suctioning the nose and mouth/throat is to remove fluid in the upper airways, which is likely to be a mixture of amniotic fluid, oral secretions, and lung liquid? By 2010 it was well established that most lung liquid cleared after birth is removed across alveolar epithelia and suctioning of the upper airway is presumably unlikely to influence this.

Response: Thank you for this comment. We have revised this part of the manuscript, which now reads: “but only in 2010 [4], it was advised against routine oro- and nasopharyngeal suctioning because of concerns that harm outweighs the potential benefits.”

           b) On page 2, lines 49-50, it is not clear how “the risk of bias in non-randomised observations” would have reduced the sensitivity of the ILCOR systematic review to detect serious adverse effects. The lack of SAEs detected by the review is more likely to be due to selective reporting in the original studies, or possibly recruitment bias (the studies enrolled mostly healthy infants who were not in need of significant resuscitation).

Response: We have revised the manuscript according to the reviewer’s suggestion: “Serious adverse events were not confirmed by the randomized trials in the ILCOR review, which may be due to selective reporting in the original studies, or possibly recruitment bias as the studies enrolled mostly healthy infants who were not in need of significant resuscitation.”

  1. The abstract states that the effect of suctioning on tone and skin colour was the ‘main’ (presumably primary) outcome of the study and the rate (as a proportion) of babies receiving suctioning was a secondary outcome. It would be helpful to use the same ordering and identification of primary and secondary in the last paragraph of the introduction (which explains the hypotheses, but in reverse order).

Response: The order of primary and secondary outcomes / order of the hypotheses has been changed, please see the last lines of the section “Introduction” for the updated text. (And we have revised from using the term ‘main’ to using ‘primary’ outcome)

Methods:

  1. The study is described as prospective, but the impression from section 2.3 is that this was a retrospective analysis nested within a prospective study. Clarification would be helpful.

Response: The reviewer is correct. This has now been written clearer in the methods section: “A retrospective analysis of prospectively collected observational data.”

  1. The methods for the subgroup analysis specified in the last sentence of the introduction are not explained.

Response: The methods used for subgroup analyses have now been included in the revised methods section.

  1. Page 3 line 99 “All infants placed on the resuscitation table during the study period were recorded and assessed for eligibility….”. It would be helpful to the reader to set out the inclusion and exclusion criteria more clearly in the methods. There is an implication in the results that the only exclusions among infants placed on the resuscitation table were those whose parents opted out and those for whom the recording or transcript was uninterpretable, but is this correct? The abstract suggests that receiving positive pressure ventilation was also an inclusion criterion, but this is not clarified in the methods. The proportion of infants for whom recordings were made is only 3.3% of total births, which, if it resembles the number who were transferred to a resuscitation table, (or even the proportion receiving PPV) seems a surprisingly low proportion. Was there a high proportion of parents who opted out, or some other reason why there were so few eligible recordings?

Response: The inclusion criteria have now been described in more detail: “The inclusion criteria in this study are all infants regardless of gestational age brought to the resuscitation table undergoing stabilization and resuscitation actions including PPV and airway suctioning. Non-suctioned infants who underwent PPV are included as a control group with regards to immediate clinical outcomes.» 

Recent observational data indicate that the incidence of PPV is 3,6% (Bjorland PA, Oymar K, Ersdal HL, Rettedal SI. Incidence of newborn resuscitative interventions at birth and short-term outcomes: a regional population-based study. BMJ Paediatr Open. 2019;3(1):e000592.), similar to the 3,3% of infants in our cohort.

  1. For the primary outcome - could the authors comment on why the decision was made to use skin colour judged from video recordings as an outcome? A previous study (O'Donnell CP, et al. Clinical assessment of infant colour at delivery. Arch Dis Child Fetal Neonatal Ed. 2007 Nov;92(6):F465-7) suggests) that there is very wide variation in SpO2 levels at which different observers consider that cyanosis has resolved, when observing video recordings. Does this observation have much value at all? Was there a reason to not use Spo2 levels for the study instead of visual impressions?

Response: Part of the decision to focus on muscle tone and skin colour was driven by the completeness of available data on these variables, as opposed to pulse oximetry data that were only available for 13 infants due to technical issues and this not being part of the routine data collection for facilitated debriefings. Also, not much research exists on this topic compared with research regarding SpO2-levels and heart rate.

The study by O'Donnell CP. et al, has several differences from our study and research material. Firstly, they have inaudible sound from their data, we have all audio transcribed for all the cases in the available database. Secondly, the evaluations / determinations of skin colour judged by the 27 observers were done in varying ambient light conditions. Thirdly, they have provided research based on 20 videos, we have provided research based on 302 videos of both suctioned and non-suctioned infants. In total, this gives us the possibility to provide different conclusions and suggestions due to the number of video recordings the research was conducted from.

  1. Also, how well can muscle tone be interpreted on a video recording? Do the authors have any evidence of the validity of judgements made from observation of videos, compared to the judgement of a person caring for the baby, (who can feel resistance to passive movements, test limb recoil etc. and assess reflex responses by handling the baby)? Was it in fact only posture and responses to various kinds of stimuli that could be observed and how reliable are these observations?

Response: We have included in the revised manuscript: “good muscle tone as evaluated by flexed extremities and spontaneous movements”. We do not have evidence of the validity of our judgments, but are clear about the limitations associated with our study design.

  1. Why were other aspects of the Apgar score which may have greater reliability for reflecting the infant’s true state (such as heart rate) not assessed?

Response: All Apgar score components were assessed, but the results regarding muscle tone and skin colour were the most striking findings which we wanted to present with this manuscript. Regarding total Apgar score, please see the revised table 1 presenting the immediate outcomes of the non-suctioned infants.

  1. For the secondary outcome, I recommend that more justification is needed as to why airway suctioning before initiation of PPV was regarded as evidence of guideline non-compliance. This may reflect the Norwegian newborn resuscitation guidelines, but many international guidelines do indicate that suctioning may be appropriate in the setting of evidence of obstruction to either spontaneous or assisted ventilation. Perhaps consider explaining the exact wording of the Norwegian guidelines to help explain the use of the proxy. While other aspects of the local guidelines are outlined in section 2.1, the criteria for suctioning are not specified.

Response: The exact wording of the Norwegian guidelines to help explain the use of the proxy is now provided in the revised manuscript text. Also, we consider that it is not possible (in most instances) to diagnose airway obstruction without attempting to provide PPV first.

  1. The term ‘ad modum’ may not be familiar to many readers – although it is a Latin term that can be appropriately translated to “consistent with” or “in the manner of” it is not in very common use. I suggest replacing with a more familiar term.

Response: Point taken; the term is replaced with a more familiar term. “According to” has been used in the revised manuscript.

  1. Was use of control groups considered – e.g. observations of infants (at a comparable time point) who did not require respiratory support or did not receive suctioning?

Response: Use of control groups was considered but only infants undergoing PPV were video recorded. In the revised manuscript we have added data (revised table 1) from non-suctioned infants undergoing PPV with focus on immediate clinical outcomes including need for intubation and chest compression, subgrouped into preterm and late preterm/term infants.

Results:

  1. Page 4 line 155 – the implication is this sentence seems to be that if there was spontaneous breathing, the airway could not have been partly obstructed, or that suctioning is only appropriate in the setting of a complete obstruction. Is this implication intended? If so, is it justifiable?

Response: No, this implication was not intended and the sentence has been rephrased. Spontaneously breathing infants can have some degree of airway obstruction and also laboured breathing affecting their need for relieving / stabilizing actions to be taken to solve the ongoing respiratory problem.

  1. Page 4, section on Effect of airway suctioning on muscle tone and skin colour – new subgroup analyses that were not mentioned in the introduction or methods are described. Please state in the methods whether these analyses were decided a priori or performed post-hoc. The same applies to the sub-analyses by gestation in Tables 1 and 2.

Response: This information has now been provided in the methods section.

  1. Page 4, same section – The meaning of the statement that 19 infants were tracheally intubated, unrelated to whether suctioning was performed or not is not at clear. Please reword.

Response: The sentence has been revised as recommended. Thank you.

  1. In several places -- “who” is the usual relative pronoun used to describe human beings, not “that”.

Response: Your advice for grammar correction is taken.

  1. In Tables 1 and 2, what does “baby off crib number” mean? Please also explain the terms “normal”, “NICU” and “died or OR”? Are these categories meant to be mutually exclusive, and if so, why?

Response: Explanations of the meaning of the phrases / groups have now been provided. Baby off crib describes the timepoint “end of the resuscitation” (i.e., what happens to the infant post-resuscitation: given to mother, admitted to the NICU, or death in the operating room / delivery room). These categories are mutually exclusive. The variable “Baby off crib” has been renamed to “Infant post-resuscitation” where normal, NICU, died in DR or OR has been explained in table 1.

Discussion:

  1. Page 7, paragraph beginning line 237 – the implication is that the proportion of infants receiving suctioning are too high, but do we know what the ‘right’ rate is? Are the authors aware of any literature that reports what proportion of infants (and particularly what proportion of infants who are deemed to need some form of resuscitation) should optimally receive suctioning of the nose and throat? It is my understanding that the ILCOR treatment recommendations suggest against routine suctioning of all or most healthy infants but were careful that to acknowledge that the procedure would be sometimes required. Plugs of mucous, blood clots etc. (and in MSAF-exposed infants, clumps of meconium) can sometimes obstruct the airway, and these may not be revealed until the procedure is performed. As alluded to in the discussion, in an infant who is too preterm or too depressed to swallow effectively, even clear fluid could obstruct the airway. As indicated in the paragraph, the NRP course indicates airway suctioning as one of the potential “MR SOPA” ventilation-corrective steps. Before intubation, one would assume it is reasonable to suction a higher proportion of infants than those found to have something causing obstruction, on the basis that it would be less forgivable to miss a true obstruction.

Response: Thank you for your insights. We agree that suctioning should be done when required in situations where plugs of mucus, blood clots, blood, secretions, lung liquid, and/or clumps of meconium obstruct the airway. However, an obstructed airway can also occur due to inappropriate positioning, decreased airway tone and/or laryngeal adduction, especially in preterm infants at birth. Suctioning should only be done when initial PPV attempts fail, as part of MR SOPA – or before intubation. Thus, the incidence of airway suctioning is high in the submitted manuscript. PPV must be prioritized and is recommended before suctioning even in infants born through meconium-stained amniotic fluid. Also, which evidence do we have, before mask-bag ventilation has been attempted, to know if the airway is obstructed? Therefore, PPV should be the first action.

  1. Following from the previous point, it is hard to discern whether the authors considered the rate (as a proportion) of suctioning in their study was optimal, too high, or too low, especially since they report that a third of infants improved after the procedure. Is it possible to draw any conclusion from this study? Regarding safety, without any data on heart rate or saturation levels, is it possible to draw any sound conclusions about safety from this study? What gap in knowledge does the study fill?

Response: The findings in the submitted manuscript are understandably controversial, and not quite in agreement with what the current resuscitation guidelines recommend. Our intention is not to provide evidence to change the guidelines, merely to present the observation that many neonates became more vital as an immediate response to suctioning, i.e., it seems that oro-pharyngeal suctioning functioned as an effective stimulus without causing reflex bradycardia and apnoea. The data show that suctioning of the upper airway occurs too frequently, but without clear evidence of harm.

We have concluded that the rate of airway suction in this study is too high and the presented data confirms our hypothesis that airway suction in neonatal resuscitation and stabilization situations in our hospital (AUH) seems to be performed “routinely”.

The knowledge gaps filled by this study is more availability of data regarding premature infants, and vigorous infants being airway suctioned and not undergoing airway suction. Immediate outcomes are presented for preterm and late preterm/term infants, and clinical and to some extent immediate physiological outcomes of untimely / incorrectly performed airway suctioning. As previously described, we had only a very low number of pulse oximetry-registrations, the reason for which we did not present pulse oximetry-data in the manuscript.

  1. Page 7, line 240, the ILCOR scoping review would have been superseded by the 2022 systematic review, so may not require citation. The consensus on gaps in knowledge identified by the systematic review should be readily available in the Consensus on Science with Treatment Recommendations from the systematic review (reference 4).

Response: Thank You very much for your suggestion, advice taken.

References:

  1. References 5 and 9 require correction of formatting and abbreviation style, and some of the references that are only available on-line have missing URL data. 

Response: The mentioned references are now corrected. Thank You.

Reviewer 2 Report

This is a very interesting study by Berisha et al evaluating the changes in muscle tone and skin color following airway suctioning in infants receiving delivery room positive pressure ventilation

·      Line 164&165: ‘Stimulated a vigorous cry in 47 (29.5%) infants’- Please mention the distribution of vigorous and non-vigorous infants in both preterm and near-term groups and their response to airway suctioning in a table. 

·      Line 167: Please clarify how you got ‘35 (22%) infants that were admitted to NICU’ as you mentioned there were 27 <33 weeks GA babies admitted to NICU and 62 ≥ 33 weeks babies admitted to NICU.

·      Table 2: Is there a reason for so high number of NICU admissions (57.8%) in the term and near-term infant category as shown in table 2?

·      Line 164- Page 5: You have mentioned that the suction led to stimulation- this could be a correlation but difficult to find a causal relation based on the study design. Could this be the result of other measures being done during resuscitation or the course the infant would have taken irrespective of the fact whether suction was done or not? It is very difficult to derive causality in this study design. 

·      Can you please clarify if the suctioning was oral suctioning or deep suctioning?

·      Table 1: Please mention what was your definition of normal vs abnormal heart rate.

·      Can you mention the number of infants that got intubated or received chest compressions after suctioning.

·      What was the rationale to choose <33 weeks GA and ≥ 33weeks as the cut-off to compare preterm and term babies?

·      Do you have the data for the babies that did not get suctioning to compare the outcomes with the group that got suctioning? 

·      Was there any reason why suction was done in vigorous babies? 

·      Page6- line 219-220: How can you derive the conclusion that the chest compression was a direct result or not from suctioning based on the study design?

Author Response

Dear Editor of Children and the reviewers – thank you for valuable comments and suggestions for improvement of our original paper entitled “Retrospectively assessed muscle tone and skin colour following airway suctioning in video-recorded infants receiving delivery room positive pressure ventilation” (note: title revised after review).

We have revised the paper according to the suggestions and made a point-by-point description of the changes as well as responses/clarifications in the following. We hope that the manuscript has improved.

Comments and Suggestions for Authors

This is a very interesting study by Berisha et al evaluating the changes in muscle tone and skin color following airway suctioning in infants receiving delivery room positive pressure ventilation

·      Line 164&165: ‘Stimulated a vigorous cry in 47 (29.5%) infants’- Please mention the distribution of vigorous and non-vigorous infants in both preterm and near-term groups and their response to airway suctioning in a table.

Response: Thank you for this comment. Table 3, a new table, presents the response of airway suctioning in both groups (vigorous and non-vigorous infants).

  • Line 167: Please clarify how you got ‘35 (22%) infants that were admitted to NICU’ as you mentioned there were 27 <33 weeks GA babies admitted to NICU and 62 ≥ 33 weeks babies admitted to NICU.

Response: Thank you for this comment. The 35 (22%) infants that were admitted to NICU, are the ones which were non-vigorous from birth and had persistent inadequate spontaneous respiration when suction alone, but also suctioning combined with PPV and tactile stimulation was performed with the result being no improvement of muscle tone. These 35 infants are among the ones who were admitted to NICU and are found in both GA groups with the total number of infants admitted to NICU being 89 (27+62 infants). Table 3 (a new table) now presents these results better grouped according to GA <33 weeks and >33 weeks and if they were vigorous/non-vigorous from birth.

  • Table 2: Is there a reason for so high number of NICU admissions (57.8%) in the term and near-term infant category as shown in table 2?

Response: Our cohort includes only ventilated babies (i.e., a selected population), and many infants were moderately asphyxiated. Most NICU admissions of term infants were due to inadequate respiration and the on-call doctors wanting to stay on the safe side after having provided delivery room stabilizing measures. The general admission rate of infants to neonatal intensive care units (NICU) in Norway is just below 10%. Note: The initial “Table 2” has been incorporated into “Table 1: Characteristics and immediate outcomes of suctioned (n=159) and non-suctioned (n=143) preterm and late preterm/term infants who also received PPV”.

  • Line 164- Page 5: You have mentioned that the suction led to stimulation- this could be a correlation but difficult to find a causal relation based on the study design. Could this be the result of other measures being done during resuscitation or the course the infant would have taken irrespective of the fact whether suction was done or not? It is very difficult to derive causality in this study design.

Response: It could be that other measures having been done before or are done after the suction episode(s) lead to a stimuli such as crying and/or movement of limbs. Based on the available data, we are able to evaluate if a single suction episode or repeated suction episodes are the reason for why a stimuli was seen when suction is superficial, deep, or combined superficial plus deep. Also, the data clearly shows that the improvement of crying/muscle tone was seen immediately after the suction episode indicating a relationship between the two.

  • Can you please clarify if the suctioning was oral suctioning or deep suctioning?

Response: Please see table 2, where we have added a row with the number and percentage of infants undergoing superficial (oral, nose, mouth), deep (trachea, glottis, larynx/ vocal cords), and cases where superficial and deep suction has been combined.

  • Table 1: Please mention what was your definition of normal vs abnormal heart rate.

Response: Definition of normal vs abnormal heart rate in table 1 is specified. Thank you for your comment.

  • Can you mention the number of infants that got intubated or received chest compressions after suctioning.

Response: Thank you for this question. The presented data are the number of infants intubated after having been suctioned and/or received chest compression – however, four of the intubated infants were not intubated in immediate timely relationship to airway suctioning.

  • What was the rationale to choose <33 weeks GA and ≥ 33weeks as the cut-off to compare preterm and term babies?

Response: This was a pragmatic decision to ensure a reasonable number of infants in the preterm category. According to the revised Australian pregnancy care guidelines and risk of preterm birth, the “average gestational age for all preterm births was 33.3 weeks”.

  • Do you have the data for the babies that did not get suctioning to compare the outcomes with the group that got suctioning?

Response: Yes, these are now presented, please see the revised table 1 “Table 1: Characteristics and immediate outcomes of suctioned (n=159) and non-suctioned (n=143) preterm and late preterm/term infants who also received PPV”.

  • Was there any reason why suction was done in vigorous babies?

Response: No good reason is found in the available data as to why suction was performed besides that the majority of the suction episodes were performed late during the resuscitation and stabilization of the infant(s), perhaps because the team / the doctor leading the team wanted to be sure that there was no amniotic fluid/ mucus causing any respiratory problem. We speculate that airway suction in these cases is due to a bad habit such as routine suctioning of most / all infants just to “be sure”.

  • Page6- line 219-220: How can you derive the conclusion that the chest compression was a direct result or not from suctioning based on the study design?

Response: Thank you for this question. It is difficult to conclude that the chest compression was or was not a direct result from suctioning or non-suctioning, but we use the word ‘probably’ when discussing this finding. After having reviewed the transcripts multiple times, we conclude that ineffective ventilation may result in a need for chest compression, in addition to severe perinatal asphyxia (the cohort included 5 infants that developed HIE and went on to be cooled). In some cases the chest compression was also initiated before ventilation and airway suction had been performed at all, while in the majority of the cases chest compression was initiated after several minutes of ventilation.